# Improving the Professional Level of Managers Through Individualized Recommendation to Enhance the Quality of Air Pollutant Management in China

**Xia Xiao [1,2], Hanwen Qin [1], Huijuan Fu [1] and Chengde Zhang [1,*]**

[1] School of Information and Safety Engineering, Zhongnan University of Economics and Law, Wuhan 430073, China; xiaoxiaandxue@163.com (X.X.); zuel_hanwenqin@163.com (H.Q.); gracefhj@126.com (H.F.)

[2] Laboratory Management Center, Wuhan Qingchuan University, Wuhan 430204, China

\* Correspondence: chengdezhang@zuel.edu.cn

**Abstract:** With the rapid development of the economy, and fossil fuel consumption lacking systematic emission controls, China has experienced substantially elevated concentrations of air pollutants, which not only degrades regional air quality but also poses significant impacts on public health. However, faced with the demand for a large number of experts in air pollution protection, people with real expertise for air pollutant management are difficult to find. Therefore, individualized recommendation is an effective and sustainable method for enhancing the professional level of managers and is good for improving the quality of air pollutant management. Thus, this paper initially proposes a novel framework to recommend strengths in air pollutant management. This framework comprises four stages: data preprocessing is the first stage; then, after constructing ability classifications and ability assessment strategies, activity experiences are transformed into corresponding ability values; next, a multilayer perceptron deep neural network (MLP-DNN) is used to predict potential types according to their ability values; finally, a hybrid system is constructed to recommend suitable and sustainable potential managers for air pollutant management. The experiments indicate that the proposed method can assess the full picture of people's strengths, which can recommend suggestions for building a scientific and rational specialties recommendation system for governments and schools. This method can have significant effects on pollutant emission reduction by enhancing the professional level of managers with regard to air pollutant management.

**Keywords:** air pollutants; health risk; sustainable management; recommendation

## 1. Introduction

Air pollution in China has attracted considerable attention from the public, scientists, and policymakers [1,2], as the hazards of ultrafine particles affect human health. Air pollution can directly or indirectly affect human health, causing physical discomfort, leading to disease or even death [3]. Regarding air pollution and management, current research is focused on four parts: sources [4–8], relationships with health [9–12], extreme events [13–17], and control methods [18–23]. For example, Ashbaugh [4] used statistical methods to categorize air pollution sources in the United States. Qiu et al. [6] showed that emissions from motor vehicles are one of the main sources of air pollution. Li et al. [7] found that China's air pollution is correlated with rapid industrialization. Cong [9] found that ambient air pollution from waste gas emissions was associated with multiple cancer incidences in Shanghai in a retrospective population-based study. Zhang et al. [14] used remote sensing data and numerical simulations to analyze a heavy air pollution event in Chengdu and found the sources and causes of this event. Lawrence et al. [16] conducted research on air pollution control engineering

and obtained some air pollution control models through an empirical analysis. Through the above studies, it has been found that research on air pollution mainly focuses on the source, methods, cases, and relationships between industry and pollution. Relatively less research has been carried out on the relationship between air pollution and control and the resulting pollution control. In addition, studies on these topics have always attracted public attention [24]. Improving the quality of managers is also crucial for air pollution control. The uneven professional accomplishment of managers is also an important reason for the ineffectiveness of air pollution control in China. Few studies have been conducted on the methods of improving the quality of managers, especially for China.

With the popularity of the internet and smartphones, people are exposed to considerable air pollutant management information every day. China needs thousands of professional people to monitor air pollution problems every year. However, it is difficult for people to determine their specialties about air pollutant management. Therefore, this paper attempts to find a new method for fully analyzing the characteristics of their specialties in relation to air pollutant management. Generally, universities are an important source of professional training. This paper can help freshmen make professional choices that correspond better with their expertise. Thus, more talented people will have more opportunities to become key members in air pollutant management in the future. However, in China, the reform of college students' enrollment system is thriving. Since 2018, most of the universities in China have begun to recruit students without distinction between majors, which affects nearly 1,000,000 students each year. Thus, it is urgent to recommend that people in air pollutant management work continually improve their level of air pollutant management. In the future, this new policy could influence the approach taken by many universities [25]. This would affect millions of students each year. It would significantly promote the reform of the university education system in the future [26]. Additionally, it would broaden students' choices and enrich the connotation of university education in the future. In the short term, it could help students adapt to society's demand for diversified talent [27]. Over time, it could further strengthen the teaching and research level of higher education institutions, improving the educational strength and research ability of China [28]. Enrollment without specific classification of a major has exerted a significant influence on the university student cultivation system [29]. Universities must make timely adjustments in cultivating personnel and create a reasonable and customized talent-cultivating model for students in the context of recruiting. The cultivation of talent is becoming a much-discussed issue: for example, Yin Hui proposed a structured cultivating framework based on multidisciplinary intersection and a comprehensive and distinctive curriculum system to form the curriculum for students [30]. Zhigang Liu proposed the "1+2+1" electronic information undergraduate talent training model theory and gave the configuration of general courses and major courses for numerous students [31]. Jiang Youwen proposed a new cultivation model to organize students' behavior and professional choice ability, attempting to stimulate students' motivation to choose their own profession and optimize the problem of professional diversion in the process of talent cultivation according to the AMO (Ability, Motivation, and Opportunity) theory [32]. AMO theory is an important basic theory in the field of behavioral science. Its core idea is that the behavior and performance of an individual is a function of ability, motivation, and opportunity. Whether an individual will take an action depends not only on whether the individual has the ability to take an action but also on whether the individual has the motivation or expectation of taking an action. Additionally, it also depends on where the individual is. The objective environment is the influence of opportunity factors. The decrease in psychological value in any of the three dimensions of ability, motivation, and opportunity may lead to a decrease in individual behavior and performance level. However, all of the research is about optimizing the structure of courses. There is no research using the student perspective to explore students' individualized cultivation methods. Regarding the issue of student development, enrollment problems can be easily detected [33]. (1) The time for students to access professional courses has been postponed. Large-category enrollment has broadened the knowledge scale of students; however, the time for students to access professional knowledge has been postponed accordingly. Therefore, the time spent

studying professional courses is reduced; meanwhile, some of those courses are in constant change, which might affect the future employment of students [34]. (2) In colleges, students are cultivated in an undifferentiated educational mode, which makes it difficult for schools to discover the students' strengths or weaknesses and provide individualized guidance [35,36]. (3) Cultivating feedback is lagging. Under large-category cultivation, the school only focuses closely on students from a higher level and with more granularity [37–39]. However, this could lead to a disconnection between school education and student feedback. The students are unsure about their abilities, and the school knows nothing about their students' abilities either. This has made a significant impact on the full development of students [40].

Air pollution management in China is badly in need of a large number of professionals every year, and talent in this area needs a long time of professional training to suit future jobs. However, the enrollment of large categories is not professional. This would reduce professional learning for one and a half years. Thus, it is very difficult to master professional knowledge in a short time. Therefore, this requires that students in this major have certain talent in air pollution management. However, in China, most freshmen do not know what they are good at. Thus, there is an urgent need for a set of models that can be analyzed and recommended according to students' daily data. To solve the problem that air pollution management poses needs a large number of professionals, while a large number of students who are gifted in air pollution do not know the correct way to choose a profession and professional knowledge. This paper tries to find an effective solution to this problem.

This paper proposes a novel individualized recommendation framework for air pollutant management level improvement. First, a large number of data are collected anonymously. Second, a new ability evaluation model is established. Third, multilayer perceptron deep neural network (MLP-DNN) is used to predict the cultivating types of people. Finally, cultivating types and ability information are integrated to personally recommend everyone.

The main contributions of this paper are summarized as follows:

(1)　Aiming to accurately recommend talent for each person from numerous data, MLP-DNN classification information, user-based information, and content-based information are integrated to generate individualized recommendations, which accurately and efficiently learn abilities about different kinds of individuals. Moreover, by analyzing students' first-view data, we can more objectively identify their talents and reduce the influence of subjective factors, which would provide considerable help for air pollutant management level improvement.

(2)　Faced with the chaotic and irregular mass of survey data, a multidimensional ability evaluation model is proposed to acquire the abilities and talents of different people in different aspects. It formulates targeted countermeasures for improving the level of managers by finding people with talent in the field of air pollutant management.

(3)　Both user-based and content-based information are taken as important information. They are combined in a hybrid way to recommend suitable and sustainable potential managers for air pollutant management.

The main content of this paper is organized as follows. The proposed framework is discussed in Section 2. The experiments and results are presented in Section 3. Section 4 concludes the paper.

Some definitions in this paper are given as follows.

- *Ability assessment strategy*: A set of methods is used to evaluate students' daily performance for all abilities. It is composed of an experience-oriented set, result-oriented set, and bonus rule. More specifically, the ability assessment strategy has the following characteristics. (1) It has a clear view of the items that can be regarded as bonus items. (2) It gives an absolute credit value that can be precisely calculated.
- *Cultivating type*: This refers to the method in which the students are recommended to progress during their undergraduate education; for instance, "academic type" is aimed at cultivating

students to become academic talents, and "design type" aims to cultivate students to become design talents.

- *Cultivating content*: The activities or competitions that students have to experience for educational reasons, such as reading books, doing homework, watching courses online, and participating in competitions.

- *Experience-based ability growth*: This refers to activities in which students participate to improve a certain aspect of their abilities [40,41]. This paper refers to the set of activities that meet the conditions as the experience-oriented set, which is denoted by *EC*, where $EC = \{c_1, c_2, c_3, \ldots c_i, \ldots, c_n\}$, and $c_i$ indicates activity. Any $c_i$ corresponds to a bonus $p_i$, and the bonus rule for experience-based ability growth is represented by $BR_{EC}$, where $BR_{EC} = \{\overline{EC}, \{p_1, p_2, p_3, \ldots, p_i, \ldots, p_n\}\}$.

- *Results-based ability growth*: This refers to a student's participation in a competition to certify a certain ability. In this paper, the set of competitions that meet the conditions is called the results-oriented set, which is denoted by *R*, where $RC = \{c_1, c_2, c_3, \ldots, c_j, \ldots, c_m\}$, and $c_j$ denotes the competition. Any $c_j$ corresponds to a bonus point $p_j$, and the bonus rule for results-based ability growth is represented by $BR_{RC}$, where $BR_{RC} = \{\overline{RC}, \{p_1, p_2, p_3, \ldots, p_j, \ldots, p_m\}\}$.

## 2. Methods

As shown in Figure 1, the proposed framework is composed of four stages: data, evaluation model, MLP-DNN, and recommendation. (1) Data. We collected data from 4328 questionnaires. (2) Evaluation model. This paper formulates nine abilities, which are scored and counted through an ability assessment strategy. (3) MLP-DNN. MLP-DNN is used to predict cultivating types for each student. (4) Recommendation. According to the predicted results of MLP-DNN, both user-based and content-based information are combined to construct a recommendation system, recommending suitable information for each student.

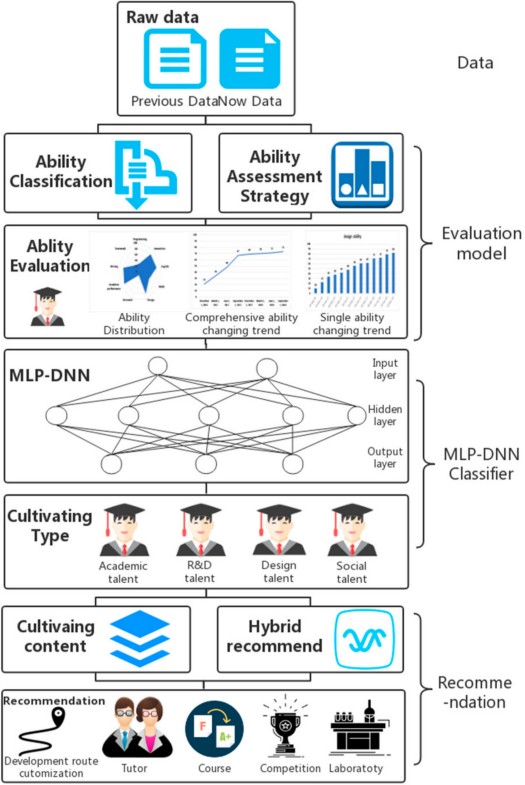

**Figure 1.** Framework.

## 2.1. Evaluation Model

We innovatively created a student ability evaluation process. To facilitate the evaluation of students' abilities, this paper classifies the abilities of students into nine categories. For the ability classification, we have nine abilities: computer technology, design ability, English ability, mathematics ability, scientific research ability, writing ability, innovation ability, academic performance, and cooperative ability. Table 1 below provides detailed information on ability classification.

**Table 1.** Classification description.

| Single Ability | Ability Description |
| --- | --- |
| Computer technology | Measures the performance of students in understanding and writing code |
| Design ability | Measures the ability of students to present in design thinking, multimedia design and implementation |
| English ability | Measures students' English learning level and English application level |
| Mathematical ability | Measures the ability of students in basic mathematics and applied mathematics |
| Scientific research ability | Measures students' interest, potential and objective strength in the direction of scientific research |
| Writing ability | Measures the ability of students in language organization, logical expression (focusing on technology) |
| Innovation ability | Measures student thinking creativity |
| Cooperation ability | Measures the awareness, ability, and results of students seeking to work with others in their studies and life |
| Academic performance | Measures the learning performance of students in basic school courses |

We innovatively propose an ability assessment strategy. This paper originally divides ability growth into two categories: experience and results, making an assessment strategy for each one. When conducting the ability assessment strategy, the student ability evaluation result is represented by $T$, where $T = \{t_1, t_2, \ldots, t_i, \ldots, t_9\}$, $t_i$ represents one of the nine abilities. The evaluation of ability is composed of three steps. (1) Input the student experience set (S) and process the element S in turn. (2) Classify the experience S into the correct ability classification. Then, the experience-oriented collection (EC) set and the result-oriented collection (RC) set are compared to determine the classification of the experience. Next, query the corresponding bonus rule $BR_{EC}$ and $BR_{RC}$ and determine the bonus score of the experience [42,43]. The above process is repeated until all the experience elements in the set S are scored and complete. (3) Ability result T is obtained. To further analyze the ability evaluation result, multilayer analysis of comprehensive ability T is carried out from the following three perspectives. (1) Comprehensive ability changing trend. The comprehensive ability of students gradually grows with changes in experience, which can reflect the growth of students more macroscopically. (2) Ability distribution. Students have their own strengths and weaknesses, and the ability distribution analysis can more intuitively show the students' abilities, which is conducive to students' strengths and weaknesses. (3) Single ability changing trend. Observing the single ability changing trend independently from the time dimension can more specifically reflect the growth of students' single abilities, to deepen the understanding of students. Through the above trials, the scores of every students' abilities are obtained.

## 2.2. MLP-DNN Classifier

MLP is a supervised learning algorithm that learns a function $f(\cdot) : R^m \rightarrow R^o$ by training on a dataset [43,44], where $m$ is the number of dimensions for input and $O$ is the number of dimensions for output. Given a set of features $X = x_1, x_2, \ldots, x_n$ and target $y$, where $x_n$ is a certain single ability score of a student, and $y$ is a cultivating type, it can learn a nonlinear function approximate for either classification or regression. MLP-DNN is used to classify the types for all students.

As shown in Figure 2, the process of MLP-DNN is composed of three steps. (1) Dataset preparation. This paper consults with experts and professors in universities to identify students and determine their future cultivating types. The cultivating type is finally classified into four types: "academic type", "R&D type" (short for research and development), "design type", and "social type". (2) Training the MLP-DNN model. A four hidden layers MLP-DNN is built. In the hidden layers, the hyperbolic tan function is taken as the activation function [45], as shown in Equation (1). In the output layer, the softmax function is taken as the activation function, as shown in Equation (2). (3) Classification.

The types of students are classified into four types: "academic type", "R&D type", "design type", and "social type".

$$g(z) = \frac{e^z - e^{-z}}{e^z + e^{-z}}, \tag{1}$$

$$softmax(z)_i = \frac{e^{z_i}}{\sum_{l=1}^{k} e^{z_l}}, \tag{2}$$

where $z$ is the output of the previous layer. In the output layer, we use the softmax function [45], as shown in Equation (2). $z_i$ is the $i$-th element of the input to softmax, which corresponds to class $i$ and $K$ is the number of classes. The loss function for our model is cross-entropy, which is given as follows.

$$Loss(\hat{y}, y, W) = -yln\hat{y} - (1-y)ln(1-\hat{y}) + \alpha\|W\|_2^2, \tag{3}$$

where $y$ represents the target value, $\hat{y}$ is the parameter estimation of y, $\alpha\|W\|_2^2$ is an L2-regularization term that penalizes complex models, and $\alpha > 0$ is a nonnegative hyperparameter that controls the magnitude of the penalty. The limited-memory-Broyden–Fletcher–Golfarb–Shanno (L-BFGS) [46,47] algorithm is used to perform parameter updates for our model training process. More details about the experiments are shown in Section 3.

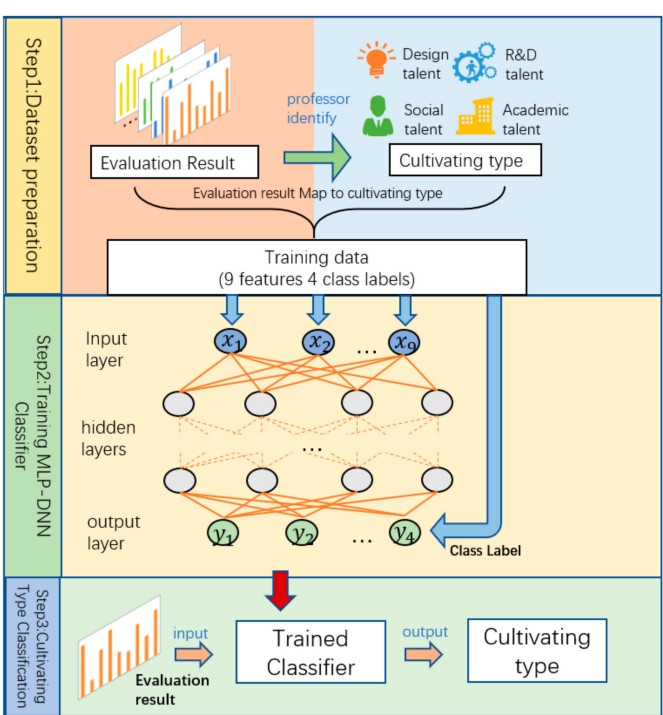

**Figure 2.** Multilayer perceptron deep neural network (MLP-DNN).

For further individualized recommendations, we build a hybrid system that combines user-based CF and content-based recommendations [48]. As shown in Figure 3, our method is composed of three stages.

*2.3. Recommendation*

First, for the input part, we prepare three main matrixes. Initially, the student evaluation result matrix is composed of evaluation results of students who share the same cultivating type that is classified by MLP-DNN at the previous stage. Each row of the matrix is defined as $\left\{t_1^{s_j}, t_2^{s_j}, \ldots, t_i^{s_j}, \ldots t_9^{s_j}\right\}$ where $S$ indicates student and $t_i^{s_j}$ represents the $i$-th ability value of the $j$-th student. Then, the content ability requirement matrix is cultivated. It is common that different activities or competitions have

different ability requirements for participants. We take advantage of this idea and build a matrix that contains the ability requirements of well-known activities or competitions that are beneficial to students' growth and development. Each row of the matrix is defined as $\left\{t_1^{d_j}, t_2^{d_j}, \ldots, t_i^{d_j}, \ldots t_9^{d_j}\right\}$, where $d$ indicates cultivating content and $t_i^{d_j}$ is the required ability value for $i$-th ability of $j$-th cultivating content. Finally, the cultivating content feedback matrix is built. The feedback value is set between 0 and 5. Each row of the matrix is defined as $\left\{f_1^{s_j}, f_2^{s_j}, \ldots, f_i^{s_j}, \ldots, f_n^{s_j}\right\}$ where $S$ indicates the student and $f_i^{s_j}$ is the feedback credit value for the $i$-th cultivating content of the $j$-th student.

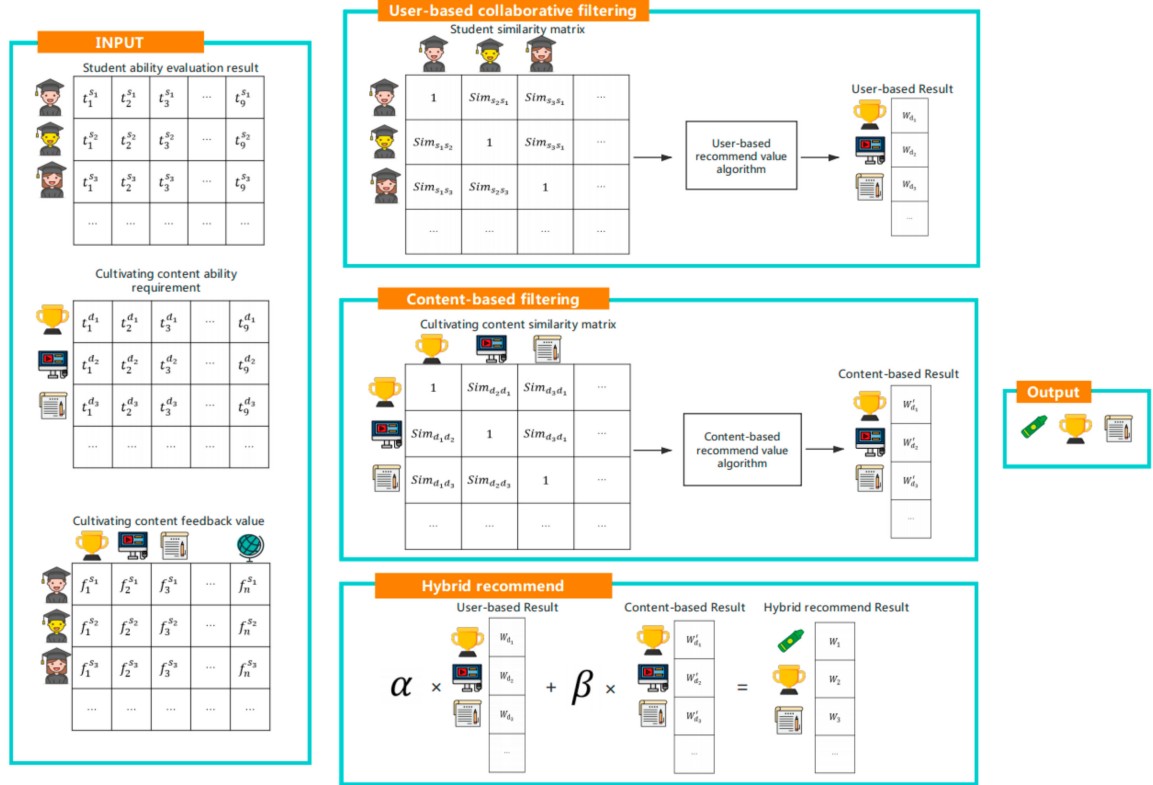

**Figure 3.** Individual recommendation.

Second, for the user-based CF, we initially build the student similarity matrix. The Euclidean distance is taken for similarity calculation, which is defined by Equations (4) and (5).

$$O.dist = \sqrt{\left(t_1 - t_1'\right)^2 + \left(t_2 - t_2'\right)^2 +, \ldots, + \left(t_9 - t_9'\right)^2}, \tag{4}$$

$$Sim = \frac{1}{1 + O.dist}, \tag{5}$$

where $O.dist$ is the Euclidean distance of the different student's abilities, $t, t'$ is the ability value of the students, $Sim$ is the similarity of the student's abilities, and $Sim$ is the value of 0–1. The larger the value, the more similar the student's abilities are. When the student similarity matrix is built, the recommended values of each cultivating content are calculated by following Equation (6).

$$W_{d_m} = \frac{\sum_{j=1}^{n}\left(Sim_j \times f_m^{S_j}\right)}{\sum_{i=1}^{n} Sim_i}, \tag{6}$$

where $W_{d_m}$ is the recommended value, $n$ is the number of students currently in the same cultivating type, $f$ is the feedback value, $j, i$ represents the student number, and $Sim$ is the similarity. For the content-based recommendation, we initially build the cultivating content similarity matrix. The Euclidean distance is taken for the similarity calculation defined in Equations (4) and (5). After building the similarity matrix, we use a content-based recommended algorithm to calculate the recommended values by Equation (7).

$$W'_{d_m} = \frac{\sum_{j=\alpha}^{n} \left( Sim_{d_m d_n} \times f_n \right)}{\sum_{i=\alpha}^{n} Sim_{d_m d_n}}, \tag{7}$$

where $W'_{d_m}$ is the recommended value, $d_n$ is the cultivating content that has been experienced, $d_m$ is the present cultivating content, $f_n$ is the corresponding feedback values, and $Sim$ is the similarity between the cultivating content.

Finally, user-based CF results and content-based recommendation results are combined for final output by adding weights to calculate the final recommended values, defined as Equation (8).

$$W = \alpha W_{d_m} + \beta W'_{d_m}, \tag{8}$$

where $W$ is the final recommended value, and $W_{d_m}$ and $W'_{d_m}$ are the respective recommended values calculated by the previous two methods, $\alpha$ and $\beta$ are the two recommended weights, which can be customized according to personal preferences. This paper regards user-based CF results and content-based recommendation results as equally important, where $\alpha = \beta = 0.5$.

## 3. Results and Discussion

### 3.1. Dataset

A total of 4328 questionnaires were distributed in the survey. The percentages of men and women in the survey were 42.88% and 57.12%, respectively. The gender ratio of the sample was reasonable, and the average age of the respondents was 18 years. Descriptive statistics of the data, rural household registration accounted for 36.99%, nonrural household registrations accounted for 63.01%; age at 18 years old students accounted for 69.83% of questionnaires.

### 3.2. Evaluation

The students' experiences were input, and the ability evaluation results were calculated. For the comprehensive ability calculation, we used the following Equations (9) and (10):

$$E = \frac{1}{9} \sum_{i=1}^{9} e_i w(e_i), \tag{9}$$

$$w(e_i) = \frac{e_i}{S}, \tag{10}$$

where $E$ is the comprehensive ability score, $e$ represents each score, and $S$ is the total score of comprehensive ability.

As shown in Figure 4, it indicates that different students show different strengths. In other words, each student has his/her own strengths in one aspect.

We selected students A, B, C, and D from our real student dataset and performed our evaluation process. Students A, B, C, and D were selected from the real student dataset, and all of them represented a certain type of student. As shown in Figure 5, there were three aspects of abilities: changing trend of comprehensive abilities, distribution of abilities, and changing trend of single ability. For the comprehensive ability changing trend in Figure 5a, there were significant differences in the starting point, the increasing range of unit time, and the increasing range of different time periods for different students' comprehensive abilities. Student A had a higher initial ability value. Each stage had a

slower ability to grow; Student B had a lower initial ability value, but the subsequent stage had a rapid increase in ability. Student C's ability had a high initial value, and the ability growth was stable during the growth phase. Student D had the ability to rise rapidly in the initial stage, but with less ability to increase in the later period. The growth of the comprehensive ability of different students was closely related to their experience. The ability distribution in Figure 5b indicates the distribution of different students in single abilities. Student A performed well in academic performance, research, and innovation. Student B had outstanding performance in mathematics and programming. Student C had obvious advantages in design, English, and innovation ability. Student D had excellent ability in teamwork, writing, and English. This indicates that the student's ability distribution is a personalized characteristic of the students. For the single ability changing trend in Figure 5c, there are great differences in upload time for different single abilities. Different single abilities are reflected in different students, with different growth rates, and the time period for the explosion and growth of single abilities is also different. This reflects the growth characteristics of single abilities of different students.

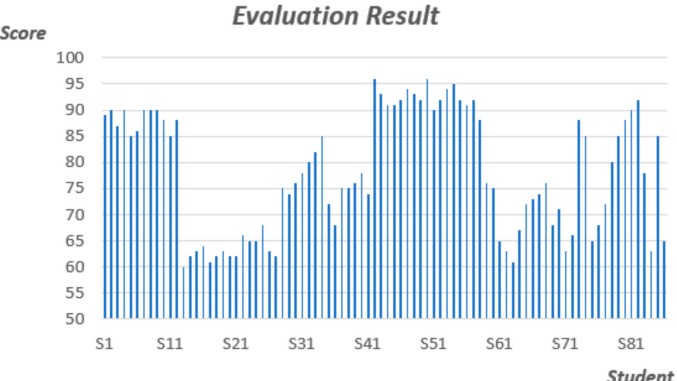

**Figure 4.** Evaluation.

*3.3. MLP-DNN*

Datasets were split into a training set and a testing set; 75% of the dataset was randomly selected as the training set, while the rest were taken as the testing set. To better evaluate the performance of the MLP-DNN model, we used the precision (P), recall (R), and F1 (F1—measure), which are defined in Equations (11)–(13), respectively [49].

$$Precision = \frac{|B_i^+|}{|A_i|},\tag{11}$$

$$Recall = \frac{|B_i^+|}{|B_i|},\tag{12}$$

$$F1 = \frac{2 \times Precision \times Recall}{Precision + Recall},\tag{13}$$

where $B_i^+$ is the number of correctly classified students of the i-th cultivating type, $A_i$ is the number of students that were classified to the i-th cultivating type by MLP-DNN, and $B_i$ is the number of students that were the real i-th cultivating type referring to the real dataset. As $F_1$ considers both *precision* and *recall* values, it was mainly used to evaluate the performance.

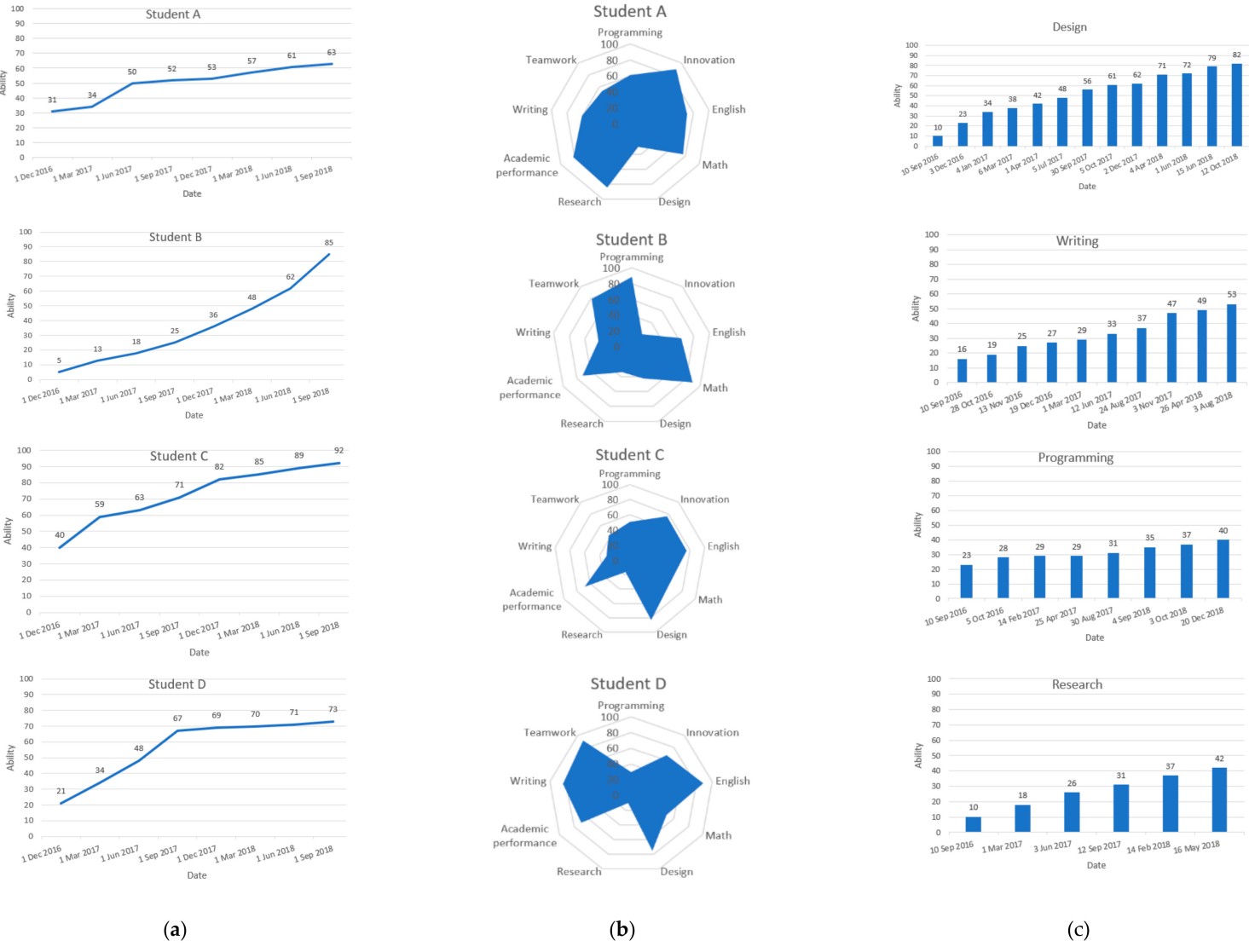

**Figure 5.** Results of multilayer analysis: (**a**) comprehensive ability changing trend; (**b**) ability distribution; (**c**) single ability changing trend.

As shown in Figure 6, it can be seen that MLP-DNN achieves a result with precision of 0.97 and recall of 0.96. To make a comparison and confirm the classification effect, the comparison results are shown in Figure 6. MLP-DNN yields better results than other methods, such as naïve Bayes, KNN, logistic regression, and SVM.

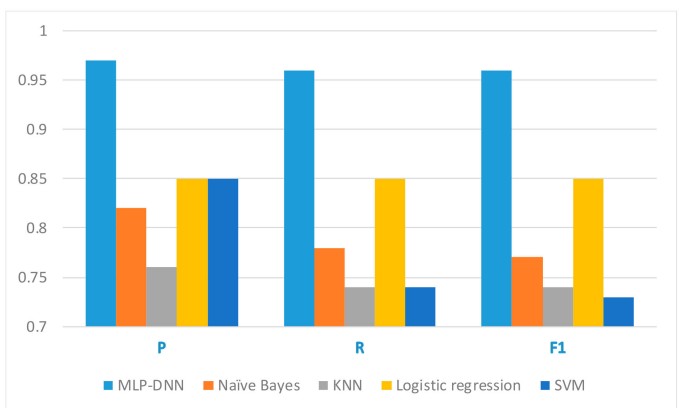

**Figure 6.** Comparison.

### 3.4. Recommendation

We randomly selected 50 students who were classified as different types for evaluation. First, we conducted our model and produced the recommendation results for each surveyed student, and we selected the top 10 recommendation results from our method. Second, we asked students to give a score ranging from 1–5 to every recommendation result, where '1' represented 'useless', '2' represented 'not so good', '3' represented 'not bad', '4' represented 'helpful' and '5' represented 'great help'. Last, based on those feedback scores, we introduced mean absolute error (MAE) to test the recommendation effects [50]. It is defined as Equation (14).

$$MAE = \frac{1}{m} \sum_{i=1}^{m} |f_i - y_i|,$$ (14)

where $f_i$ is the student score value, and $y_i$ is our method recommended value. The final result is shown in Figure 7.

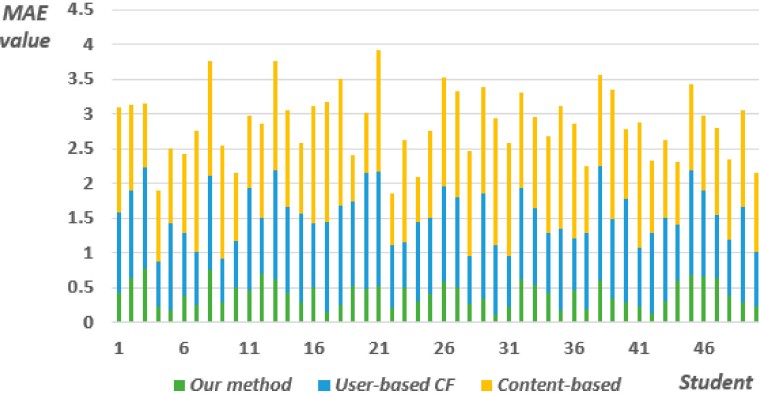

**Figure 7.** Mean absolute error (MAE) results comparison.

As shown in Figure 7, our method outperforms the user-based CF and content-based recommendation. Moreover, the MAE is relatively low among the tested students. Some of them even reached below 0.1, which shows the results are very close to the students' real preferences. In summary, the results revealed that our prediction is much greater for students' tastes.

As shown in Figure 8, it is very encouraging that 47% of students considered our recommendation result of great help to them. It is proven that our method can provide a good recommendation for students without a major to develop their abilities.

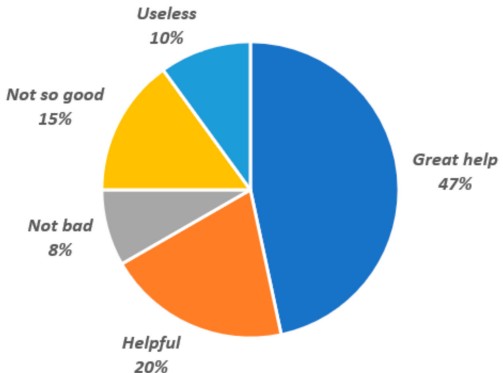

**Figure 8.** Subjective evaluation.

## 4. Conclusions

This paper finds a new mode for improving air pollution management in addition to several methods, such as sources, relationships with health, extreme events, and control methods. Compared with the previous models, the method proposed in this paper is based on long-term and sustainable air pollution management, which may not be effective in the short term. However, it establishes a series of personnel training mechanisms from talent discovery, talent cultivation to talent recommendation. Currently, people are facing the rapid development and impact of big data and artificial intelligence. How to find talented individuals who are truly suitable for air pollution management from the vast crowd of people is very important. In the near future, China will experience the largest enrollment model reform in history, involving millions of people and hundreds of colleges and universities every year, including more than 100 colleges and thousands of students for air pollution management majors alone. The new student training model proposed in this paper will play an important role in the improvement of the overall educational level of China and have a far-reaching impact on the talent training model. Since China has just started the reform of enrollment education for one year, there are fewer experimental subjects. In the future, we will extend our method to more students and more majors for experiment and promotion.

**Author Contributions:** C.Z. organized this study and conducted the study design, performed the statistical analysis, and drafted the manuscript. X.X. contributed to the study design, interpretation of the analysis, and revision of the manuscript. H.Q. and H.F. contributed to prepared datasets, performed the statistical analysis, and drafted the manuscript. All read and approved the final manuscript.

**Funding:** This work was supported by "the Fundamental Research Funds for the Central Universities", Zhongnan University of Economics and Law (Grant Number: 2722019PY062), "the Humanities and Social Sciences Research Project of Hubei Education Department" (Grant Number: 18G012), "the Program for Excellent Project of Student Work in Colleges and Universities of Hubei Province" (Grant Number: 2018XGJPB3017), "the Laboratory Research Projects of Colleges and Universities in Hubei Province" (Grant Number: HBSY2018-48), "the Fundamental Research Funds for the Central Universities", Zhongnan University of Economics and Law (Grant Number: 2722019JCT037), and "the Ministry of Education, Humanities, Social Science Project of China" (Grant No. 19A10520035) and "the Fundamental Research Funds for the Central Universities".

**Acknowledgments:** The authors would like to express their deep gratitude to their teammates, including Zini Bie and Dandan Jin.

**Conflicts of Interest:** The authors declare no conflict of interest.

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
