# Peer review of "Improving the Professional Level of Managers Through Individualized Recommendation to Enhance the Quality of Air Pollutant Management in China"

_sustainability, doi:10.3390/su11216094_

Round 1
Reviewer 1 Report
The author/s elaborate an article proposed a novel framework to recommend strengths in air pollutants management
The structures and contents of the manuscript should be further reorganized and improved, respectively.
The introduction must provide an extensive overview of recent developments in this specific area that fall within the scope of the journal and not only a list of published studies.
The originality and significance of the paper needs to be further clarified.
The objectives of this study should be clearly stated to present the novelty of the study in the Introduction.
The methodology is explained more clearly.
In the Conclusion, please discuss the implications of your research. Conclusion should be revised and go deeper, it would be more interesting if the authors focus more on the significance of their findings regarding the importance of the interrelationship between the obtained results and the real application.
Author Response
We’d like to thank associate editor and reviewers for their valuable comments and constructive suggestions to improve the quality of this manuscript. In this revised version, we have considered the comments and suggestions very cautiously and seriously, making sure our new manuscript has addressed each concerned issue properly. ( Attached please find the revised version for this manuscript)
Point 1: The structures and contents of the manuscript should be further reorganized and improved, respectively.. 

Response 1: Thanks for pointing out it. We appreciate your high-standard review on our paper to make it better. After careful consideration, we reorganize the structure of this manuscript according to the steps of the proposed framework. In this way, it would be easy for readers to follow the ideas of this manuscript. The title of each step is consistent with the steps in the framework. Some unnecessary data (2.1 Materials Collection) are put into the experimental part (3.1 Dataset). Some unnecessary graphs have been deleted (Figure 2 Materials collection analysis).
According to the suggested comments, the content of this manuscript is whole improved. Considering that the students are not continuous data, we redrawn Figures 4 and 7 to transform the line graph into a clustering bar graph.
Point 2: The introduction must provide an extensive overview of recent developments in this specific area that fall within the scope of the journal and not only a list of published studies.
Response 2: Thank you for pointing out it. We have revised the introduction part to provide a clear description of the overview of recent developments in this specific area. (Line 36 to line 48)
Point 3: The originality and significance of the paper needs to be further clarified.
Response 3: Thank you for pointing out it. We agree that it will make our originality and contributions outstanding if we further clarified them. So, we have revised this part to provide a clear description to state the original contributions of our paper. (Line 114 to line 126)
Point 4: The objectives of this study should be clearly stated to present the novelty of the study in the Introduction.
Response 4: Thank you for pointing out it. We agree that it will make our original contributions outstanding if the objectives of this study are clearly stated. So, a new paragraph has been added to state the objectives of our paper. (Line 97 to line 107)
Point 5: The methodology is explained more clearly.
Response 5: We have carefully checked all the symbols, equations and models in section 2 and added the explanation about those symbols, equations and models, which are not clearly explained in the previous manuscript. In addition, we give more corresponding explanation about the ability assessment strategy. We have added more details for ability classification.
Point 6: In the Conclusion, please discuss the implications of your research. Conclusion should be revised and go deeper, it would be more interesting if the authors focus more on the significance of their findings regarding the importance of the interrelationship between the obtained results and the real application..
Response 6: Thanks for your suggestion. We have revised the conclusion part to provide a deep discussion about the implications about our research. More details about the importance of the interrelationship between our research and the air pollution management improvement are added in the paragraph of conclusion. (Line 330 to line 344)
Reviewer 2 Report
1) Figure 2 is unnecessary.
2) A more detail description of student classification types should be added.
3) Figure 5 should be presented in better resolution.
4) Figure 5, figure 8 - Graphs represent a discrete phenomena (students are not continuous set), so it should not be presented as a line graph. I suggest some king of a bar graphs to be used.
5) When talking about results of student A, student B, student C, student D it is not clear where is the origin of their data. Were they selected from real students' dataset, are they just some representative proxy data, ...?
6) 267-270 + Figure 7, it is not clear if this is authors' own result or cited result.
7) The Precision and Recall function definition should be improved to be more understandable.
8) The student test should be described in greater detail - how it was performed, what kind of tests were performed, etc.
Author Response
We’d like to thank associate editor and reviewers for their valuable comments and constructive suggestions to improve the quality of this manuscript. In this revised version, we have considered the comments and suggestions very cautiously and seriously, making sure our new manuscript has addressed each concerned issue properly. (Attached please find the revised version for this manuscript.)
Point 1: Figure 2 is unnecessary. 

Response 1: Thanks for pointing it out. Figure 2 is deleted.
Point 2: A more detail description of student classification types should be added.
Response 2: Thank you for pointing out it. More details about the description of ability classification types are added. (Line 141 to 145)
We mainly focus on 9 abilities: computer technology, design ability, English ability, mathematics ability, scientific research ability, writing ability, innovation ability, academic performance and Cooperation ability. Table1 below gives a detailed information of ability classification.
Table1.Ability classification description
|
Single ability |
Ability description |
|
Computer tenology |
Measuring the performance of students in code understanding and code writing |
|
Design ability |
It mainly measures the ability of students to present in design thinking, multimedia design and implementation. |
|
English ability |
Measuring students' English learning level and English application level |
|
Mathematical ability |
Measuring the Ability of students in basic mathematics and applied mathematics |
|
Scientific research ability |
Measuring students' interest, potential and objective strength in the direction of scientific research. |
|
Writing ability |
Measuring the ability of students in language organization, logical expression (focusing on technology). |
|
Innovation ability |
Measuring student thinking creativity |
|
Cooperation Ability |
Measuring the awareness, ability, and results of students seeking to work with others in their studies and life. |
|
Academic performance |
Measuring the learning performance of students in School Basic Courses |
Point 3: Figure 5 should be presented in better resolution.
Response 3: Thanks for pointing it out. I have modified the figure into bar graph. Considering the large number of test students, we use cluster bar chart to demonstrate the result of the evaluation. (Line 231 to 233).
Point 4: Figure 5, figure 8 - Graphs represent a discrete phenomena (students are not continuous set), so it should not be presented as a line graph. I suggest some king of a bar graphs to be used.
Response 4: Thank you for pointing it out. For figure 8, I have changed the line graph into the bar graph. For figure 5, I have taken your advice and use bar graph as a better resolution, and it was modified at previous response. (Line 282 to 284)
Point 5: When talking about results of student A, student B, student C, student D it is not clear where is the origin of their data. Were they selected from real students' dataset, are they just some representative proxy data, ...?.
Response 5: Thanks for pointing it out. Student A,B,C,D are selected from the real students’ dataset, where each of them represent one type of students. In addition, MLP-DNN is used to predict four types of students which are summarized in the way of student A,B,C and D.
More details about student A,B,C and D are added (Line 236-238) as follows:
We selected student A,B,C and D from our real student dataset and perform our evaluation process. The student A,B,C and D are selected from the real student dataset, and all of them can represent a certain type of students.
Point 6: 267-270 + Figure 7, it is not clear if this is authors' own result or cited result.
Response 6: We use real data as the input of MLP-DNN, and get the classification results, which are our own experimental results.
Point 7: The Precision and Recall function definition should be improved to be more understandable.
Response 7: Thank you for pointing it out. The descriptions of the Precision and Recall functions are improved for more understandable. (Line 262 to 265)
where is the number of correctly classified students of i th cultivating type, is the number of the students which have been classified to the i th cultivating type by MLP-DNN ,and is the number of students which are real i th cultivating type referring to the real dataset. Since considers both and values, it is mainly used to evaluate the performance.
Point 8: The student test should be described in greater detail - how it was performed, what kind of tests were performed, etc.
Response 8: Thank you very much for your suggestion. Actually, when we evaluated the performance of the recommendation result, we asked some students to do a satisfaction survey. Firstly, we conducted our method and produce the recommendation results for each surveyed students, and we pick up top 10 recommendation results from our method; Secondly, we asked student to give a score ranging from 1-5 to every recommendation result, where 1 represent ‘useless’, 2 represent ‘not so good’, 3 represent ‘not bad’, 4 represent ‘helpful’ and 5 represent ‘great help’. Lastly, based on those feedback scores, we used the MAE method to evaluate the effectiveness of proposed method. (More details are shown as follows: (line 274 to 280)
We randomly select 50 students who have been classified at different types for evaluation. Firstly, we conducted our model and produced the recommendation results for each surveyed students, and we picked up the top 10 recommendation results from our method. Secondly, we asked student to give a score ranging from 1-5 to every recommendation result, where ‘1’ represent ‘useless’, ‘2’ represent ‘not so good’, ‘3’ represent ‘not bad’, ‘4’ represent ‘helpful’ and ‘5’ represent ‘great help’. Lastly, based on those feedback scores, we introduce Mean Absolute Error (MAE) to test the recommendation effects [50]. It is defined as equation (14).

Round 2
Reviewer 1 Report
The authors have greatly improved the article and have followed the guidelines.
For this reason, the article is for me to be published